# Differential Disruption of Glucose and Lipid Metabolism Induced by Phthalates in Human Hepatocytes and White Adipocytes

**DOI:** 10.3390/toxics12030214

**Published:** 2024-03-14

**Authors:** Yaru Tian, Miao Xu, Hailin Shang, Lijuan You, Jing Yang, Xudong Jia, Hui Yang, Yongning Wu, Xingfen Yang, Yi Wan

**Affiliations:** 1Food Safety and Health Research Center, School of Public Health, Southern Medical University, Guangzhou 510515, China; tianyarusmile@163.com; 2NHC Key Laboratory of Food Safety Risk Assessment, China National Center for Food Safety Risk Assessment, Beijing 100021, China; jiaxudong@cfsa.net.cn (X.J.); yanghui@cfsa.net.cn (H.Y.); 3Department of Clinical Nutrition, West China Hospital, Sichuan University, Chengdu 610044, China; xumiao1726@wchscu.cn; 4Laboratory for Earth Surface Processes, College of Urban and Environmental Sciences, Peking University, Beijing 100871, China; shanghailin@stu.pku.edu.cn (H.S.); wany@urban.pku.edu.cn (Y.W.); 5School of Public Health, Shandong Second Medical University, Weifang 261053, China; 20220846@stu.wfmc.edu.cn (L.Y.); 20220848@stu.wfmc.edu.cn (J.Y.)

**Keywords:** phthalates, metabolism, hepatocytes, adipocytes, PPARs

## Abstract

Phthalic acid esters (PAEs), commonly used as plasticizers, are pervasive in the environment, leading to widespread human exposure. The association between phthalate exposure and metabolic disorders has been increasingly recognized, yet the precise biological mechanisms are not well-defined. In this study, we explored the effects of monoethylhexyl phthalate (MEHP) and monocyclohexyl phthalate (MCHP) on glucose and lipid metabolism in human hepatocytes and adipocytes. In hepatocytes, MEHP and MCHP were observed to enhance lipid uptake and accumulation in a dose-responsive manner, along with upregulating genes involved in lipid biosynthesis. Transcriptomic analysis indicated a broader impact of MEHP on hepatic gene expression relative to MCHP, but MCHP particularly promoted the expression of the gluconeogenesis key enzymes G6PC and FBP1. In adipocytes, MEHP and MCHP both increased lipid droplet formation, mimicking the effects of the Peroxisome proliferator-activated receptor γ (PPARγ) agonist rosiglitazone (Rosi). Transcriptomic analysis revealed that MEHP predominantly altered fatty acid metabolism pathways in mature adipocytes (MA), whereas MCHP exhibited less impact. Metabolic perturbations from MEHP and MCHP demonstrate shared activation of the PPARs pathway in hepatocytes and adipocytes, but the cell-type discrepancy might be attributed to the differential expression of PPARγ. Our results indicate that MEHP and MCHP disrupt glucose and lipid homeostasis in human liver and adipose through mechanisms that involve the PPAR and adenosine monophosphate-activated protein kinase (AMPK) signaling pathways, highlighting the nuanced cellular responses to these environmental contaminants.

## 1. Introduction

Phthalates, a family of chemical compounds known as PAEs, serve as plasticizers to enhance the pliability, clarity, durability, and lifespan of plastic materials, predominantly in polyvinyl chloride (PVC). Ubiquitous in consumer goods, they appear in items ranging from food packaging and toys to medical apparatus and personal care products [1,2]. The ubiquity of PAEs raises concerns due to their leaching propensity, which allows them to enter the human body via ingestion, inhalation, or dermal absorption [3,4]. Di(2-ethylhexyl) phthalate (DEHP), as the most-used plasticizer, correlates with a spectrum of deleterious health effects, including endocrine disruption, reproductive toxicity, and potential carcinogenicity [5,6]. Given these health implications, DEHP usage faces stringent regulations, especially in children’s products, medical devices, and food contact materials. Regulatory bodies and health organizations are actively evaluating the risks of DEHP to safeguard public health [7].

Dicyclohexyl phthalate (DCHP) is a chemical substance used as a plasticizer and considered to be a lower toxicity substitute for DEHP. The DCHP presence in environmental and consumer contexts has also been documented in water sources, foods, indoor dust, and numerous other products [8]. Human exposure to DCHP, and its metabolite MCHP, has been confirmed through their detection in various biological samples [9,10]. Notably, certain populations show high urinary MCHP levels, frequently exceeding 1.5 µg/L [11]. Additionally, DCHP has been identified in the cord blood samples from pregnant women, with means concentrations around 125 µg/L [12]. Despite these human exposure data, the research on the health impact of DCHP remains sparse, highlighting a critical area for scientific inquiry. Therefore, DCHP has been proposed by the U.S. Environmental Protection Agency as a high-priority substance for risk evaluation [13]. 

Although reproductive impact has been commonly used as a reference endpoint for the tolerable daily intake (TDI) estimation of PAEs, the metabolic systems for some phthalates could be more sensitive compared to their reproductive toxicity [14]. For example, animal studies indicated that DCHP may trigger intestinal pregnane X receptor (PXR) activation, potentially elevating cardiovascular disease risk through hypercholesterolemia and ceramide production [15]. Emerging research also links phthalate human exposure to metabolic disorders such as obesity, insulin resistance, hepatotoxicity, and cardiovascular diseases [16,17,18,19]. However, the underlying mechanisms for PAE-associated metabolic disorder are still poorly understood. Particularly, the differential disruption of each PAE with similar chemical structures on glucose and lipid metabolism is not clear, necessitating further investigation to corroborate these associations. 

In this study, we established a 3D-spheroid model of hepatocytes and a human white adipocytes differentiation model, which could determine the cellular metabolic changes with high-content imaging. These cell models were applied to investigate the impact of the metabolites of the widely used phthalate, DEHP, and DCHP on glucose and lipid metabolism. MEHP and MCHP were found to exhibit differential disruptions of glucose and lipid metabolism in human hepatocytes and white adipocytes. The differential effects between the two cell types are likely due to the varying expression of PPARs. The results provide new insights into potential metabolic health consequences induced by different metabolites of phthalates. 

## 2. Materials and Methods

### 2.1. Chemicals and Reagents

DEHP (AccuStandard), MEHP (AccuStandard), DCHP (AccuStandard), MCHP (AccuStandard) were purchased from AccuStandard (New Haven, CT, USA), with a purity of >98%. Calcium pantothenate (Sigma Aldrich, St. Louis, MO, USA), Human insulin (MCE), Biotin (Sigma Aldrich), Dexamethasone (Sigma Aldrich), Indomethacin (Sigma Aldrich), Isobutylmethylxanthine (Sigma Aldrich), 3,3′,5-Triiodo-L-thyronine (T3, Sigma Aldrich) were purchased from Sigma Aldrich (Shanghai, China) or MCE (Monmouth Junction, NJ, USA). Except for calcium pantothenate (unlabeled), the purity of reagents mentioned above was greater than 95%. GW6471 (MCE), GSK3787 (MCE), T0070907 (T007, R&D), and Rosi (Cayman) were purchased from MCE (Monmouth Junction, NJ, USA) or Cayman (Ann Arbor, MI, USA). Detailed information of the reagents is provided in Appendix A.

### 2.2. Cell Culture

HepG2 cells derived from human liver hepatocellular carcinoma were sourced from the Cell Bank of Type Culture Collection Committee of the Chinese Academy of Sciences. The cells were cultured in Eagle’s Minimum Essential Medium (EMEM, ATCC, Cat: 30-2003, Manassas, VA, USA) supplemented with 10% (*v*/*v*) fetal bovine serum (FBS, Thermo Scientific, Cat: 10091148, New York, NY, USA) and 1% (*v*/*v*) penicillin/streptomycin (P/S, Thermo Scientific, Cat: 15140122, New York, NY, USA). White fat progenitor cells of human origin were obtained from the American Type Culture Collection (ATCC) and were cultured in Dulbecco’s Modified Eagle Medium (DMEM, Gibco, Cat: 11995065, New York, NY, USA) supplemented with 10% (*v*/*v*) FBS and 1% (*v*/*v*) P/S. All cells were maintained in an atmosphere of 37 °C, 5% carbon dioxide (CO_2_), and 95% humidity.

### 2.3. 3D Spheroid Cell Culture 

The HepG2 cells were seeded in a spherical microporous 96−well plate with Ultra-Low Attachment surface (Corning, Cat: 4520, Kennebunk, ME, USA), with 500 cells per well. Spheroids formed after 72 h of culture. 

### 2.4. Differentiation of Human White Fat Progenitors

Human white adipose tissue (WAT) progenitor cells (10,000 cells/well) were cultured in black-wall 96-well plates (In Vitro Scientific, Cat: 060096, Hangzhou, China) for 24 h. Subsequently, the cells were exposed to an adipogenic induction mixture in DMEM medium containing calcium pantothenate (17 μM), human insulin (0.5 μM), biotin (33 μM), dexamethasone (0.1 μM), indomethacin (30 μM), isobutylmethylxanthine (0.5 mM), T3 (2 nM), and 2% FBS for 12 days. The induction medium was refreshed every 3 days until collection. 

### 2.5. Cell Treatment

The PAEs were dissolved in Dimethyl sulfoxide (DMSO) to prepare a stock solution of 400 mM, which was stored at −20 °C. The stock solution was further diluted 1000−fold to prepare a working solution, which was subsequently diluted 2-fold to prepare PAEs ranging from 0 to 200 μM. HepG2 cells (15,000 cells/well) and WAT cells (10,000 cells/well) were seeded in 96−well plates with black walls and incubated for 24 h. Subsequently, they were exposed to DEHP, MEHP, DCHP, and MCHP. A concentration range of 0−200 μM of PAEs was employed, with all concentrations showing no cytotoxicity towards the cells. MEHP and MCHP have been widely detected in human blood, with detectable concentrations ranging from 0.007 to 1.868 μM (equivalent to 1.84−520 ng/mL) [20]. In this study, 0.07 μM of PAE was observed as benchmark dose (BMDL_10_) in the obtained dose–response curve, which was comparable to those reported in human blood. Given that the exposure duration in in vitro cell experiments is notably shorter than real human exposure periods, our research also incorporated higher treatment concentrations. Dose-response experiments for selective PPAR ligands involved various concentrations of GW6471 (0−10 μM), GSK3787 (0−10 μM), and T007 (0−2 μM). Changes in lipid uptake and accumulation in the cells can be detected as early as 24 h into treatment. However, transcription of genes into proteins takes a certain amount of time, so our processing time was 72 h. There was a minimum of 2 replicate wells for each concentration in each independent experiment, and all data represents at least 3 independent experiments.

HepG2 cells were treated with MEHP (100 μM) and MCHP (100 μM) for 72 h (each treatment concentration with a replicate well number of *n* = 4), and cells were collected for transcriptome sequencing. To examine the transcription-level changes of adipocytes at different stages, WAT cells (200,000 cells/well) were seeded in 6−well plates (Corning, Cat: 3516, Kennebunk, ME, USA) and incubated for 24 h. Subsequently, they were treated with MEHP (100 μM), MCHP (100 μM), Rosi (10 μM), and T007 (1 μM) in replicate wells (*n*= 3) for 72 h in complete medium or 12 days in induction medium. After the initial 24 h culture, the WAT cells were exposed to induction medium (refreshed every 3 days) for 12 days, followed by treatment with MEHP (100 μM), MCHP (100 μM), Rosi (10 μM), and T007 (1 μM) in replicate wells (*n*= 3) for 72 h in complete medium.

### 2.6. RNA Sequencing

Total RNA was isolated from cells using Trizol, and its integrity was evaluated with the Agilent 2100 Bioanalyzer (Agilent Technologies, CA, USA, 6 June 2023). The fragmented mRNA served as a template for synthesizing double-stranded cDNA using random hexamer primers. The purified double-stranded cDNA underwent screening, amplification, and purification processes to obtain the final library. Upon passing quality control, the library was sequenced using the Illumina Noveseq 6000 (Illumina, USA, 6 June 2023) in paired-end mode. Following raw data filtering, sequencing error rate checking, and GC content distribution examination, clean reads suitable for subsequent analysis were obtained. The clean reads were aligned to the reference genome using HISAT2 software (version 2.0.5, 6 June 2023) to obtain positional information on the reference genome. Gene expression levels for each sample were quantitatively analyzed using the featureCounts (version 1.5.0-p3, 6 June 2023) tool in the Subread software. Subsequently, DESeq2 (version 1.20.0, 6 June 2023) was used for significant differential gene expression analysis. The criteria for screening differential genes were set as |log2(FoldChange)| ≥ 1 and padj ≤ 0.05. The differential gene set underwent Gene Ontology (GO) enrichment analysis and Kyoto Encyclopedia of Genes and Genomes (KEGG) pathway enrichment analysis using the clusterProfiler software (version 3.8.1, 6 June 2023). Additionally, Gene Set Enrichment Analysis (GSEA) of the GO and KEGG datasets was performed using the local version of the GSEA analysis tool available at http://www.broadinstitute.org/gsea/index.jsp (accessed on 6 June 2023).

### 2.7. Live-Cell High Content Imaging

Cells were treated according to the “Cell Treatment” sections, washed with PBS (Gibco, Cat: C10010500BT, Suzhou, China), and then incubated with fluorescently labeled lipid probes for 30 min at room temperature (RT). For surface marker and probe staining, cells were incubated with fluorescence-conjugated antibodies and lipid probes in FluoroBrite DMEM (Gibco, Cat: A1896701, Grand Island, New York, NY, USA) for 30 min at RT. For intracellular protein staining, the cells were incubated with the Foxp3/Transcription Factor Staining Buffer Set (Thermo Scientific, Cat: 00-5523-00, Carlsbad, CA, USA) for 30 min at RT, followed by washing with the accompanying 1× permeabilization buffer. The cells were then incubated with the primary antibody, which was prepared in 1× permeabilization buffer, for 30 min at RT. After washing, cells were incubated with fluorescence-conjugated secondary antibodies, also prepared in 1× permeabilization buffer, for 60 min. Then the cells were incubated with Hoechst 33342 staining solution (Solarbio, Cat: C0031, Beijing, China) at RT for 5 min to label the cell nuclei. Finally, FluoroBriteTM DMEM was used to minimize background fluorescence. The fluorescent images from at least five sites per well were analyzed using CellReporterXpress Imaging and Analysis Software (version 6.5, Molecular Devices, 20 August 2023). The mean stained area (MSA) was calculated by normalizing the overall fluorescent staining cells (area and counts) to the total cells in the same image. The mean fluorescence intensity (MFI) was the average pixel intensity over all of the pixels in a spheroid. 

### 2.8. Statistical Analysis

Statistical analysis was conducted using GraphPad Pris (version 8.3.0, GraphPad, 20 August 2023), and all data represents at least three independent experiments. One-way analysis of variance (ANOVA) or two-way ANOVA followed by Bonferroni’s test was used for multiple groups’ analyses. *p* value less than 0.05 was considered statistically significant.

## 3. Results

### 3.1. Effects of DEHP, DCHP, and Their Primary Metabolites on Hepatic Lipid Metabolism

The exposure of HepG2 cells to 0−200 μM DEHP, DCHP, MEHP, and MCHP for 24 h resulted in increased lipid uptake (FL C12) and lipid accumulation (Lipid droplets, LDs), as determined by live-cell high-content imaging (HCI). These effects were more pronounced in treated cells compared to the control and exhibited a dose−response relationship. Notably, the primary metabolites MEHP and MCHP exerted a greater impact than the parent compounds, indicating their role in enhancing hepatic fatty acid uptake and lipid accumulation (Figure 1A,B). The first step in the metabolism of PAEs is hydrolysis to form monoesters. Hydrolytic enzymes may include several different carboxylesterases, including lipases [5]. PAEs can be rapidly metabolized in the body, with studies showing that porcine and bovine pancreatic cholesterol esterases can hydrolyze PAEs into monoesters in 15 min to 6 h [21]. Our research indicates that the effects of parent phthalates are lower than their metabolites, leading us to speculate that the harmful effects of PAEs primarily stem from their metabolites. Given the rapid in vivo metabolism of DEHP and DCHP, our further investigations will prioritize the metabolites MEHP and MCHP.

RNA sequencing revealed that MEHP induced a greater number of differentially expressed genes (DEGs) in HepG2 cells than MCHP. And a significant overlap of 23.7% of DEGs was shared between MEHP and MCHP (Figure 1D). The pathway enrichment analysis highlighted that MEHP predominantly affected pathways including non-alcoholic fatty liver disease (NAFLD), oxidative phosphorylation (OXPHOS), thermogenesis, cholesterol metabolism, and PPAR signaling (Figure 1E). MCHP also significantly influenced PPAR signaling and cholesterol metabolism. Additionally, MCHP uniquely modulated AMPK, cGMP-PKG, glucagon, and PI3K-AKT signaling pathways, implicating its potential role in glucose metabolism regulation (Figure 1E). 

### 3.2. Comparative Impact of MEHP and MCHP on Lipid Metabolism in HepG2 Cells

Previous studies have reported that MEHP could promote lipid accumulation in HepG2 cells through inducing fatty acid synthesis-related genes (FASN, SCD, PPARα, SREBP-1c) [22,23], but the effect of MCHP was scarcely investigated. In our study, GSEA of HepG2 transcriptomic data showed that both MEHP and MCHP significantly activated PPAR signaling, cholesterol metabolism, and OXPHOS pathways, with MEHP showing a stronger influence (*p* < 0.05) (Figure 2A). The examination of pathway-specific DEGs highlighted that both compounds consistently enhanced the expression of genes involved in lipid metabolism (PLIN2, ANGPTL4, CPT1A, etc.) (Figure 2B). Particularly, MCHP exerted a distinct effect on glucose metabolism-associated genes, most notably enriched in the AMPK signaling pathway (Appendix A). In addition, DEGs analysis indicated that MCHP notably increased the expression of FBP1 and G6PC1, which are critical enzymes in AMPK signaling for hepatic gluconeogenesis (Figure 2C), and this upregulation by MCHP was also corroborated at the protein level, with a contrast to MEHP (Figure 2D,E).

### 3.3. Effects of MEHP and MCHP on Lipid Metabolism in Human White Adipocytes

To examine the influence of plasticizers on adipose tissue energy metabolism, we utilized a high-content screening model using immortalized human white adipose precursor cells [24]. As white adipose cells differentiate and mature, there is an upsurge in the fatty acid uptake marker CD36 and LDs, whereas the adipose progenitor marker CD29 and mitochondrial count diminish (Figure 3A,B). Importantly, MEHP and MCHP notably enhanced CD36 expression and lipid accumulation in precursor adipocytes, mimicking the PPARγ agonist Rosi effect (Figure 3C). To further explore this, we analyzed the transcriptomic changes induced by these compounds in progenitor adipocytes (PA), differentiating adipocytes (DA), and MA, comparing them with Rosi and T007 to clarify PPAR signaling pathway involvement. RNA sequencing revealed that MEHP-treated MA share a larger proportion of DEGs with the Rosi-treated group (15.8%) than with the T007-treated group (3.7%), suggesting a strong association between MEHP effects and PPARγ activation in MA. Conversely, the MCHP-treated group showed minimal DEGs overlap with the MEHP, Rosi, and T007 groups, indicating distinct effects on the adipocytes (Figure 3D). KEGG pathway analysis highlighted that MEHP significantly alters signaling in MA, notably in the PPAR signaling and fatty acid metabolism pathways, suggesting a primary disruption in lipid metabolism (Figure 3E). In DA, the MEHP and MCHP effects diverged, with MCHP predominantly influencing glucose metabolism-related pathways (insulin signaling, glycerophospholipid metabolism, insulin resistance, AMPK), hinting at potential glucose metabolism disruption in DA. However, PA showed the least impact from both compounds (Figure 3E).

### 3.4. Distinct Impacts of MEHP and MCHP on Lipid Metabolism in Mature White Adipocytes

GSEA analysis indicated that MEHP significantly upregulates fatty acid metabolism and the PPAR signaling pathway (Normalized enrichment score, NES > 1.5, *p* < 0.01) in the MA, while MCHP’s effects were negligible (Figure 4A). Delving into the DEGs within these pathways, MEHP was found to substantially enhance the expression of pivotal lipid metabolism genes (FABP4, PLIN2, CD36, ANGPTL4, FASN, CPT1A, etc.) (Figure 4A). For instance, CD36 is involved in lipid uptake; by inhibiting LPL, ANGPTL4 helps to regulate the levels of triglycerides in the bloodstream; PLIN2 is essential for lipid droplet formation and stability. Moreover, MEHP’s disruption of lipid metabolism appears to concurrently interrupt glucose metabolism, as evidenced by alterations in the insulin resistance pathway (Figure 4B), an effect not observed with MCHP. However, in contrast to the impact on hepatocytes, either MEHP or MCHP did not functionally affect glucose uptake and expression of the gluconeogenesis key enzymes in the white adipocytes (Appendix A).

Comparative analysis between MEHP and MCHP demonstrated that lipid metabolism-related gene alterations were predominantly observed in the MEHP-treated group (Figure 4C). Lipid accumulation assays in DA and MA corroborated this finding (Figure 4D). In addition, MEHP dose-dependently upregulated ANGPTL4 and PLIN2 expressions in MA at the protein level, while MCHP’s influence on these proteins was considerably weaker (Figure 4E,F). Collectively, MEHP appears to exert a more potent influence than MCHP, likely disrupting lipid metabolism via the PPAR signaling pathway, which, in turn, impacts lipid assimilation, storage, and breakdown, as illustrated in Figure 4G.

### 3.5. Metabolic Perturbations from MEHP and MCHP Demonstrate Cell-Type Discrepancy

As a crucial regulator of lipid and glucose management, AMPK is essential for cellular energy equilibrium, acting as an energy sensor [25]. Our transcriptome analysis revealed that MEHP and MCHP exert divergent effects on the AMPK pathway in distinct cell lines. In HepG2 cells, MCHP reduced AMPK pathway activity (NES < −1), while MEHP markedly activated AMPK signaling in MA cells (Figure 5A). Further scrutiny of the DEGs within the AMPK pathway showed MCHP’s inhibition of genes governing glycolysis (PFKFB3), glycogen creation (GYS1), and fatty acid processing (FASN, ACACB, LIPE) in HepG2 cells. In contrast, MEHP enhances the expression of genes involved in fatty acid metabolism (CD36, LIPE, CPT1A, etc.) in MA cells (Figure 5B). These findings underscore the distinct, cell-specific influences of MEHP and MCHP on energy metabolic pathways.

Next, we validated the involvement of PPAR subtypes in MEHP’s and MCHP’s actions using LDs and PLIN2 as markers. Our results showed that MEHP and MCHP enhanced lipid droplet formation and PLIN2 expression in both HepG2 cells and white adipocytes. Moreover, the PPARα inhibitor GW6471 attenuates this effect in a dose-responsive manner, whereas the PPARγ antagonist T007 and PPARδ inhibitor GSK3783 are less effective (Figure 5D). In MA cells, GW6471 and T007 significantly reduce PLIN2 levels, mitigating MEHP’s enhancement, while GSK3787 shows a negligible effect (Figure 5E).

## 4. Discussion

Previous studies have indicated that DEHP and other phthalates have endocrine-disrupting effects, which may disturb the metabolic homeostasis of organisms and result in a range of detrimental health consequences, including NAFLD [26]. DCHP has been identified as an effective selective agonist of PXR, which may increase the risk of cardiovascular diseases in humans [15]. In our study, MEHP and MCHP promoted lipid uptake and lipid accumulation in HepG2 cells. It was reported that MEHP also promoted lipid synthesis and accumulation in rat BRL-3A hepatocytes [27], suggesting the common responses to MEHP across species. Carnitine palmitoyltransferase 1 (CPT1) is essential for fatty acid oxidation (FAO), facilitating the transport of long-chain fatty acids into mitochondria. Our previous study had shown that oral exposure to DEHP significantly elevated mitochondrial OXPHOS in the mouse hepatocytes [28]. Here, both phthalates dose-dependently increased CPT1A protein expression in HepG2 cells, with MEHP having a more substantial effect (Figure 2F). Thus, MEHP and MCHP enhance lipid metabolism-related gene expression, not only affecting cholesterol synthesis, lipid transport, and fatty acid synthesis, but also facilitating FAO in HepG2 cells. Additionally, MCHP’s influence on the expression of key gluconeogenesis enzymes suggests its role in modulating hepatic glucose metabolism (Figure 2G).

Phthalates are lipophilic and tend to accumulate in adipose tissue, an important regulator of lipids and glucose homeostasis [29]. Numerous studies have demonstrated a link between plasticizer exposure and obesity outcomes [30,31]. MEHP and MCHP had similar effects as the PPARγ agonist Rosi, which promoted CD36 expression and lipid accumulation in precursor adipocytes. This is in line with the recent study on some plasticizers (Diisononyl hexahydrophthalate, Diisononylphthalate, and Dipalmitoyl hydroxyproline) in the human SGBS adipocyte model [32]. In contrast, the PPARγ antagonist T007 markedly reduced CD36 expression and LDs in a dose-responsive manner (Figure 3C). These findings imply that MEHP and MCHP may act as PPARγ agonists, fostering lipid metabolism and storage in adipocytes [32].

The interference effects of MEHP and MCHP on metabolism were cell-specific. Our study showed that MCHP affected gluconeogenesis in HepG2 cells, but neither MEHP nor MCHP affected glucose uptake and gluconeogenesis in white adipocytes. However, it was reported that MEHP-treated 3T3-L1 mouse adipocytes exhibited significantly increased glucose uptake activity, due to the increased expression of Glut1/4 and secretion of Fibroblast growth factor 21 (FGF21) [29,33]. These inconsistent results suggest significant species difference between human and rodent in the glucose metabolism of adipocytes in response to PAEs.

PPARs, belonging to the nuclear receptor superfamily, comprise three isoforms (PPARα, PPARδ, PPARγ) acting as ligand-activated transcription factors pivotal for lipid and glucose metabolism and energy balance [34]. Numerous studies have demonstrated that PPARs might be targeted by PAE metabolites due to their chemical structure [28,35]. Notably, PPAR isoforms differ in tissue and cellular distribution. Our in vitro data indicate that HepG2 and MA cells display comparable levels of PPARα and PPARδ, whereas PPARγ is notably more expressed in MA cells than in HepG2 (Figure 5C). This differential expression may underpin the cell-specific effects of MEHP and MCHP [32]. Previous studies indicated that PAEs may also bind to other regulatory proteins, such as glucocorticoid receptor (GR) [36,37], PXR [15,38], and constitutive androstane receptor (CAR) [39]; our observations suggest that MEHP and MCHP disrupt energy metabolism in hepatocytes and adipocytes primarily via PPARα and PPARγ, with the differential expression of PPARγ likely driving the cell-type-specific responses.

Exposure to PAEs can cause multiple adverse health effects. Previous studies have indicated that exposure to PAEs may lead to energy metabolism-related diseases, such as NAFLD, atherosclerosis, and diabetes [5,15]. However, the molecular mechanisms underlying these effects remain unclear. Our study suggests that MEHP and MCHP can affect the energy metabolism of hepatic cells and white adipose cells through mechanisms that involve the PPAR and AMPK signaling pathways, thereby providing insights into the potential metabolic health consequences induced by PAEs.

## 5. Conclusions

In summary, our study reveals that the PAE metabolites MEHP and MCHP substantially interfere with glucose and lipid metabolic processes in human hepatocytes and adipocytes. Specifically, MEHP exerts a more significant influence on hepatocytes by promoting lipid accumulation and more extensively altering gene expression compared to MCHP, which predominantly escalates the expression of enzymes critical to gluconeogenesis. Additionally, both compounds induce an increased formation of LDs in adipocytes, with MEHP exerting a stronger impact on the pathways involved in fatty acid metabolism. Our findings indicate that although MEHP and MCHP both stimulate PPAR pathways, their differential metabolic effects in liver and fat cells may stem from variations in PPARγ expression levels. This research underscores the metabolic consequences of phthalate esters on human health and offers a framework for assessing the cumulative effects of phthalate ester combinations.

## Figures and Tables

**Figure 1 toxics-12-00214-f001:**
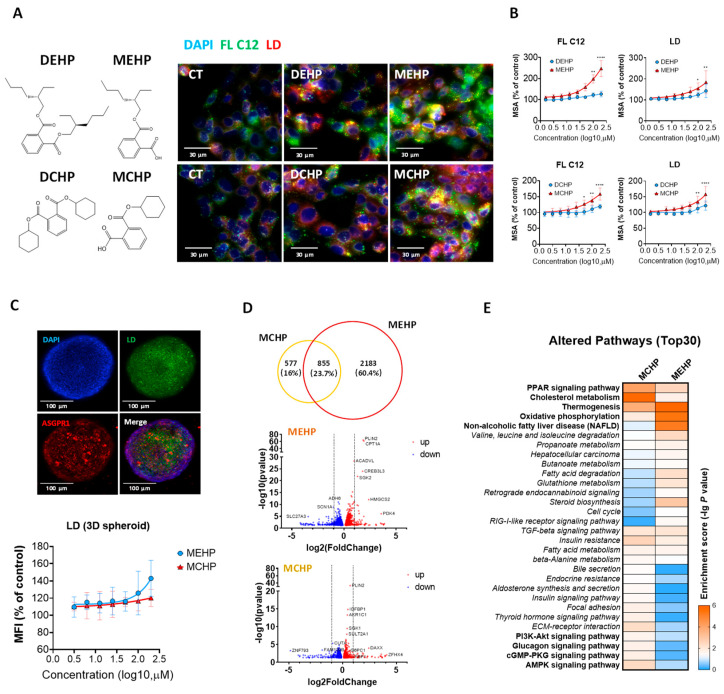
The effects of DEHP, DCHP, and metabolites on lipid metabolism in HepG2 cells. HepG2 cells were treated with different concentrations (0−200 μM) of DEHP, MEHP, DCHP, and MCHP for 24 h, and then the cellular uptake of fatty acids and intracellular lipid accumulation were detected using fluorescent probes BODIPY FL C12 and HCS LipidTOX, respectively. (**A**) Representative images of lipid uptake (FL C12, green) and lipid accumulation (LDs, red). (**B**) Dose−response curves. (**C**) Representative images of LDs in HepG2 cell 3D spheroids and dose−response curves. HepG2 cell 3D spheroids were treated with various concentrations (0−200) of MEHP and MCHP for 48 h and then incubated with the fluorescent dyes BODIPY 493/503. HepG2 cells were treated with 100 μM MEHP and 100 μM MCHP for 72 h (each treatment concentration with a replicate well number of *n =* 4), and cells were collected for transcriptome sequencing. (**D**) Venn diagram and volcano plot of DEGs in MEHP and MCHP treatment groups. (**E**) The impact of MEHP and MCHP on signaling pathways. The data were analyzed using one-way ANOVA followed by Dunnett’smultiple comparisons test. * *p*< 0.05, ** *p*< 0.01, **** *p*< 0.001 compared with the vehicle control.

**Figure 2 toxics-12-00214-f002:**
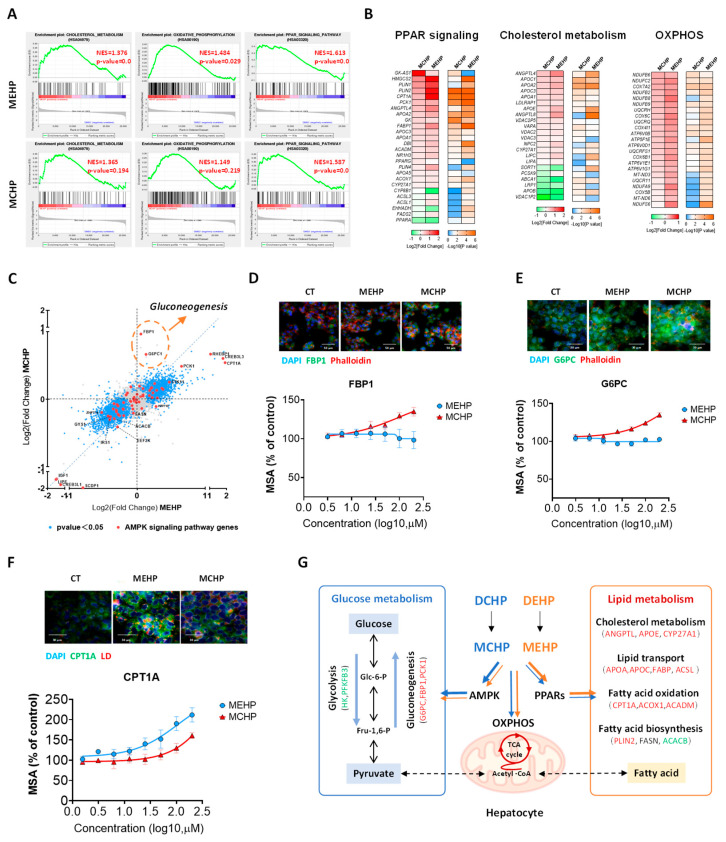
Effects of MEHP and MCHP on glucose metabolism and transcriptome in HepG2 cells. (**A**) GSEA of signaling pathways related to lipid metabolism regulated by MEHP and MCHP; (**B**) Heat map of DEGs in signaling pathways related to lipid metabolism regulated by MEHP and MCHP; (**C**) Scatter plot of DEGs in the AMPK signaling pathway under the influence of MEHP and MCHP. HepG2 cells were treated with different concentrations (0−200 μM) of MEHP and MCHP for 72 h, and the expression levels of relevant proteins were assessed. (**D**−**F**) Representative images and dose−response curves of the effects of MEHP and MCHP on the protein expression levels of FBP1, G6PC, and CPT1A in HepG2 cells; (**G**) Schematic representation of the effects of MEHP and MCHP on glucose and lipid metabolism in HepG2 cells via the AMPK, PPAR, and OXPHOS signaling pathways. Red indicates an increase in gene expression, green indicates a decrease in gene expression, and black indicates no statistically significant difference in gene expression.

**Figure 3 toxics-12-00214-f003:**
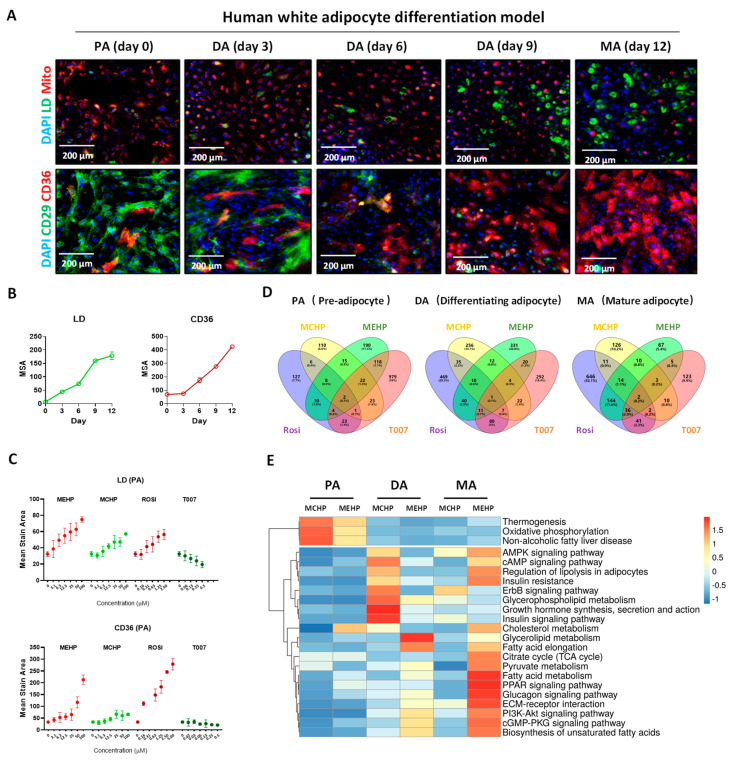
Impact of MEHP and MCHP on lipid metabolism in human white adipocytes. (**A**) Representative images of CD29, CD36, LDs, and mitochondria in PA cells at different differentiation stages. (**B**) Time−effect curve. (**C**) Dose−effect curves of lipid accumulation and CD36 expression in PA cells. (**D**) Differential gene Venn diagram of MEHP, MCHP, Rosi, and T007 treatment groups at different differentiation stages. (**E**) Impact of MEHP, MCHP, Rosi, and T007 on signaling pathways at different differentiation stages.

**Figure 4 toxics-12-00214-f004:**
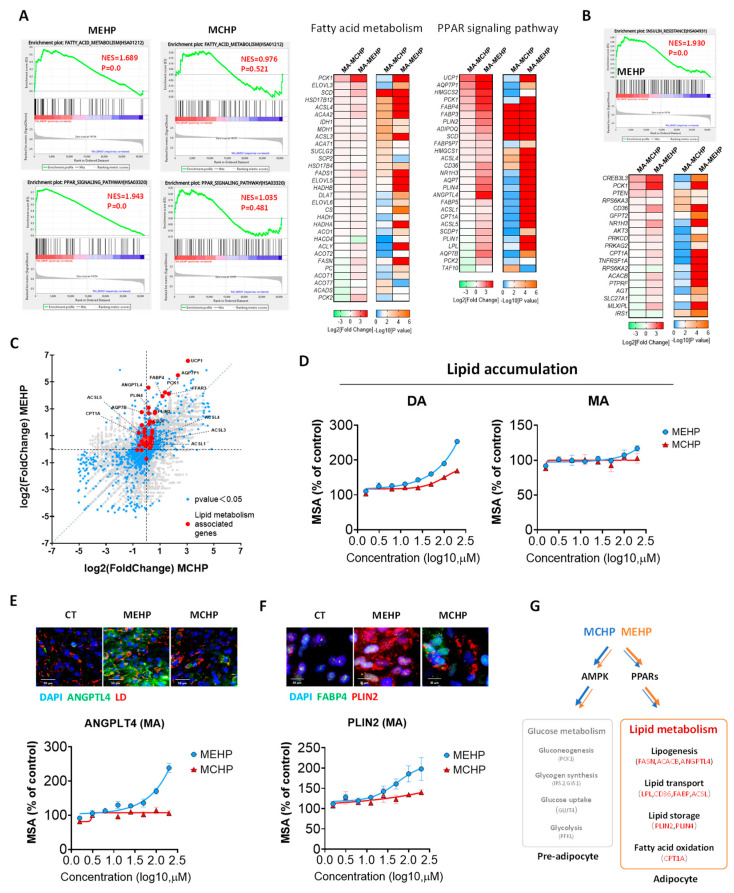
Impact comparison between MEHP and MCHP on lipid metabolism in adipocytes. (**A**) GSEA and DEGs heatmap of lipid metabolism-related signaling pathways in MA cells treated with MEHP and MCHP. (**B**) GSEA and DEGs heatmap of insulin resistance signaling pathway in MA cells treated with MEHP. (**C**) Scatter plot of changes in lipid metabolism-related DEGs in MA cells treated with MEHP and MCHP. (**D**) Dose−effect curves of lipid accumulation in PA and MA cells treated with MEHP and MCHP. (**E**,**F**) Representative images (above) and dose−effect curves (below) of the impact of MEHP and MCHP on the expression levels of lipid metabolism-related proteins ANGPTL4 and PLIN2 in MA cells. (**G**) Schematic diagram of the regulatory effects of MEHP and MCHP on the glycolipid metabolism of immature and mature adipocytes. Red indicates an increase in gene expression.

**Figure 5 toxics-12-00214-f005:**
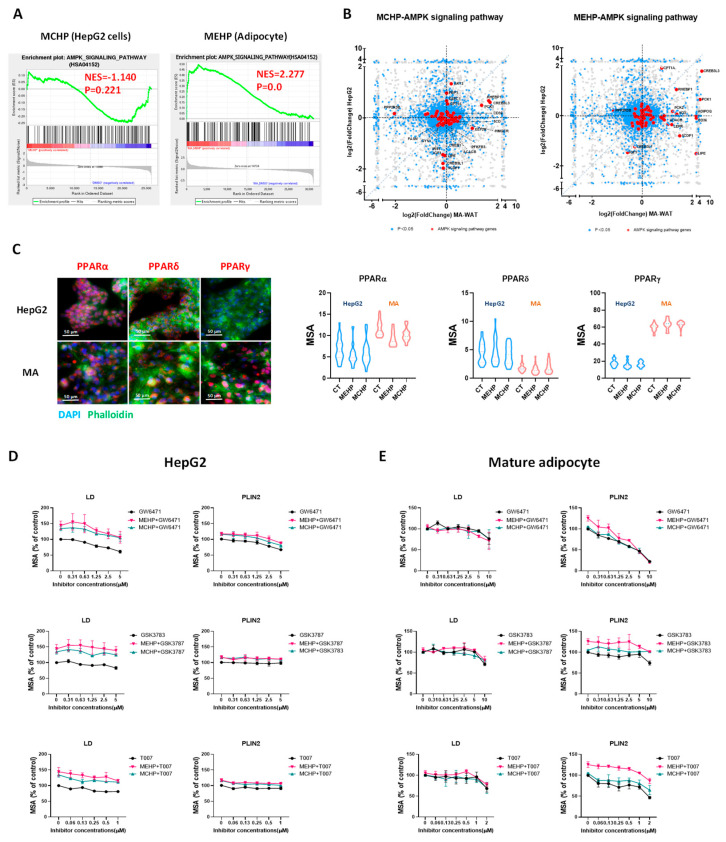
Role of PPARs in the cell-type-specific responses to MEHP and MCHP. (**A**) GSEA plot showing the regulation of AMPK signaling pathways by MEHP and MCHP in different cells. (**B**) Scatter plot of DEGs in different cell AMPK signaling pathways regulated by MEHP and MCHP. (**C**) Protein expression levels of PPARα, PPARδ, and PPARγ in HepG2 and WAT cells. MA or HepG2 cells were treated with GW6471 (0−10 μM), GSK3787 (0−10 μM), and T007 (0−2 μM) alone or in combination with MEHP (100 μM) or MCHP (100 μM) for 24 h. (**D**) Effects of MEHP or MCHP in combination with PPAR inhibitors on the expression of LDs and PLIN2 proteins in HepG2 cells. (**E**) Effects of MEHP or MCHP in combination with PPAR inhibitors on the expression of LDs and PLIN2 proteins in MA cells.

## Data Availability

Data are contained within the article and Appendix A.

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
