# Peer review of "Differential Disruption of Glucose and Lipid Metabolism Induced by Phthalates in Human Hepatocytes and White Adipocytes"

_toxics, 2024, doi:10.3390/toxics12030214_

Round 1
Reviewer 1 Report
Comments and Suggestions for Authors
Manuscript number: toxics-2842729
The authors of the manuscript examined the effects of monoethylhexyl phthalate (MEHP) and monocyclohexyl phthalate (MCHP) on glucose and lipid metabolism in human hepatocytes and adipocytes. These compounds disrupt glucose and lipid homeostasis in investigated cells through mechanisms that involve the PPAR and AMPK signaling pathways.
There are several major issues that need to be clarified:
1. I suggest adding the chemical structure of the investigated phthalates.
2. How do the results received correlate with the general human population to MEHP and MCHP ?
3. Explain in justification and in conclusion of the investigation the concentrations of phthalates used in the study in relation to the concentrations of these compounds occurring in the the environment and in human body.
4. The manuscript does not include a description of the statistical analysis.
Author Response
The Referee (I) of
Toxics:
Dear Sir:
We have received your comments on the paper entitled: Differential disruption of phthalate metabolites on glucose and lipid metabolism in human hepatocytes and white adipocytes
The authors of the manuscript examined the effects of monoethylhexyl phthalate (MEHP) and monocyclohexyl phthalate (MCHP) on glucose and lipid metabolism in human hepatocytes and adipocytes. These compounds disrupt glucose and lipid homeostasis in investigated cells through mechanisms that involve the PPAR and AMPK signaling pathways.
Thank you very much for your helpful suggestions. We appreciate your help for the improvement of the manuscript. We agree with your opinions and we have revised the manuscript according to the suggestions. Followings are our responses to your comments.
- I suggest adding the chemical structure of the investigated phthalates.
Response/Action: According to your suggestions, we have added the chemical structure of DEHP ,MEHP,DCHP and MCHP.
Figure 1A in the revised manuscript.
- How do the results received correlate with the general human population to MEHP and MCHP ?
Response/Action: According to your suggestions, we have supplemented the results received correlate with the general human population to MEHP and MCHP.
Lines 706-712:
“Exposure to PAEs can cause multiple adverse health effects. Previous studies have indicated that exposure to PAEs may lead to energy metabolism-related diseases such as non-alcoholic fatty liver disease, atherosclerosis, and diabetes[5,15]. However, the molecular mechanisms underlying these effects remain unclear. Our study suggests that MEHP and MCHP can affect the energy metabolism of hepatic cells and white adipose cells through mechanisms that involve the PPAR and AMPK signaling path-ways, thereby providing insights into the potential metabolic health consequences in-duced by PAEs.”
- Explain in justification and in conclusion of the investigation the concentrations of phthalates used in the study in relation to the concentrations of these compounds occurring in the the environment and in human body.
Response/Action: Thank you very much for your helpful suggestions.In the revised version, we elucidate the relationship between the concentrations of phthalates used in the study and the concentrations of these compounds occurring in the environment and in the human body.
Lines 169-174:
“MEHP and MCHP have been widely detected in human blood, with detectable concentrations ranging from 0.007 to 1.868 μM (equivalent to 1.84-520 ng/mL)[20]. In this study, 0.07 μM of PAEs was observed as benchmark dose (BMDL10) in the obtained dose-response curve, which were comparable to those reported in human blood. Given that the exposure duration in in vitro cell experiments is notably shorter than real human exposure periods, our research also incorporated higher treatment concentrations. ”
- The manuscript does not include a description of the statistical analysis.
Response/Action:Thanks for your reminder, according to your suggestions, we have added the statistical analysis in the revision.
Lines 297-301:
“2.8. Statistical Analysis
Statistical analysis was conducted using GraphPad Pris (version 8.3.0; GraphPad) and all data represents at least three independent experiments. One-way analysis of variance (ANOVA) or two-way ANOVA followed by Bonferroni’s test was used for multiple groups’ analysis. P value less than 0.05 was considered statistically significant.”
Sincerely,
Yongning Wu
China National Center for Food Safety Risk Assessment

Reviewer 2 Report
Comments and Suggestions for Authors
The topic is of some scientific interest.
Unfortunately, the text is confusing and difficult to read.
Overall, I believe that important changes need to be made before publication can proceed. The main problems I highlight are:
- there are numerous sentences repeated without cause (lines 120-126; 132-137; 141-146; 162-165; ...)
- acronyms should be presented when the word is first indicated in the text and then used correctly, the use of acronyms requires a check throughout the text
-line 77 to 83 is a conclusion and not an introduction
-the description of chemicals and supplies requires a verb in the second sentence and the elimination of lots
- the percentage concentration expressed as volume/volume is given (v/v) and not (vol/vol)
have been left the descriptive phrases of the template, it is to recheck the whole paper (ex: lines 171-174, 372-375, 390-392, 396-406...)
- the presentation of the results is particularly confusing and unclear. It requires a more schematic and synthetic rewriting,
- the figures are very difficult to read, they must be selected with greater criteria and properly described
-there is no "discussion" paragraph in which a comparison between the results obtained in this study and comparable literature surveys should be included
-the bibliographical references must all be re-checked: in many cases not all the authors are indicated, the number 13 is not clear to what it refers, the criteria of the journal must be followed
Author Response
The Referee (II) of
Toxics:
Dear Sir:
We have received your comments on the paper entitled: Differential disruption of phthalate metabolites on glucose and lipid metabolism in human hepatocytes and white adipocytes
The topic is of some scientific interest. Unfortunately, the text is confusing and difficult to read. Overall, I believe that important changes need to be made before publication can proceed. The main problems I highlight are:
Thank you very much for your helpful suggestions. We appreciate your help for the improvement of the manuscript. We agree with your opinions and we have revised the manuscript according to the suggestions. Followings are our responses to your comments.
1)- there are numerous sentences repeated without cause (Lines 120-126; 132-137; 141-146; 162-165; ...)
Response/Action: Thanks for this reminding. We have removed duplicate sentences and carefully checked throughout the manuscript to avoid similar mistakes in the revision.
Lines 167-168:
“A concentration range of 0-200 μM of PAEs was employed, with all concentrations showing no cytotoxicity towards the cells. ”.
Lines 196-264:
“The purified double-stranded cDNA underwent screening, amplification, and purification processes to obtain the final library”.
Lines 269-270:
“Gene expression levels for each sample were quantitatively analyzed using the featureCounts tool in the Subread software”.
Lines 286-289:
“The cells were then incubated with the primary antibody, which was prepared in 1X permeabilization buffer, for 30 minutes at RT. After washing, cells were incubated with fluorescence-conjugated secondary antibodies, also prepared in 1X permeabilization buffer, for 60 minutes. Then the cells were incubated with Hoechst 33342 staining so-lution (Solarbio, Lot: C0031) at room temperature for 5 minutes to label the cell nuclei. Finally, FluoroBriteTM DMEM was used to minimize background fluorescence”.
2) - acronyms should be presented when the word is first indicated in the text and then used correctly, the use of acronyms requires a check throughout the text
Response/Action: Thanks for your reminder. We have carefully checked the article to ensure the correct use of acronyms.
Lines 30:
“mature adipocytes (MA)”
Lines 71:
“pregnane X receptor (PXR) activation”
Lines 94:
“3,3′,5-Triiodo-L-thyronine (T3, Sigma Aldrich)”
Lines 97:
“T0070907 (T007, R&D)”
Lines 29:
“agonist rosiglitazone (Rosi)”
Lines 273:
“Kyoto Encyclopedia of Genes and Genomes (KEGG)”
Lines 350:
“differentially expressed genes (DEGs)”
Lines 690:
“Fibroblast growth factor 21 (FGF21)”
3) -Lines 77 to 83 is a conclusion and not an introduction
Response/Action: Thanks for your reminder. According to your suggestions, we have rewritten the sentences as follows.
Lines 83-86:
“MEHP and MCHP were found to exhibit differential disruptions of glucose and lipid metabolism in human hepatocytes and white adipocytes. The differential effects between the two cell types are likely due to the varying expression of Peroxisome proliferator-activated re-ceptors (PPARs).”
4) -the description of chemicals and supplies requires a verb in the second sentence and the elimination of lots
Response/Action: According to your suggestions, we have redescribed the chemicals and supplies information.
Lines 91-98:
“Calcium pantothenate (Sigma Aldrich), Human insulin (MCE), Biotin (Sigma Aldrich, CAS#: 58-85-5, Cat#:B4639-100mg), Dexamethasone (Sigma Aldrich), Indomethacin (Sigma Aldrich), Isobutylmethylxanthine (Sigma Aldrich), 3,3′,5-Triiodo-L-thyronine (T3, Sigma Aldrich) were purchased from Sigma Aldrich or MCE. Except for calcium pantothenate(unlabeled), the purity of reagents mentioned above were greater than 95%. GW6471 (MCE), GSK3787 (MCE), T0070907 (T007, R&D), and Rosi (Cayman) were purchased from MCE or Cayman. Detail information of the reagents are provided in Table S2.”
5) - the percentage concentration expressed as volume/volume is given (v/v) and not (vol/vol)
Response/Action: Thanks for your reminder. We have modified the expression according to your suggestion.
Lines 102-149:
“The cells were cultured in Eagle's Minimum Essential Medium (EMEM, ATCC, Lot: 30-2003) supplemented with 10% (v/v) fetal bovine serum (FBS, Thermo Scientific, 10091148) and 1% (v/v) penicillin/streptomycin (P/S, Thermo Scientific, 15140122). White fat progenitor cells of human origin were obtained from the American Type Culture Collection (ATCC) and were cultured in Dulbecco’s Modified Eagle Medium (DMEM, Gibco, Lot: 11995065) supplemented with 10% (v/v) FBS and 1% (v/v) P/S. All cells were maintained in an atmosphere of 37°C, 5% carbon dioxide (CO2), and 95% humidity”.
6) - have been left the descriptive phrases of the template, it is to recheck the whole paper (ex: Lines 171-174, 372-375, 390-392, 396-406...)
Response/Action: Thanks for your reminder. By incorporating the feedback from reviewer Ⅲ and Ⅳ, we have removed all templates that appeared in the article and carefully reviewed the entire manuscript to avoid similar mistakes.
7) - the presentation of the results is particularly confusing and unclear. It requires a more schematic and synthetic rewriting,
Response/Action: Thanks for your suggestion. By incorporating the feedback from reviewer Ⅲ and Ⅳ, we have separated the section of result and discussion the revised manuscript. And we have rewritten the results section more succinctly and comprehensively in the revised edition.
Lines 302-643:
Section of “Results” in the revised manuscript.
8) - the figures are very difficult to read, they must be selected with greater criteria and properly described
Response/Action: This is our mistake of uploading the lower resolution figures in the submission system. By incorporating the feedback from reviewer Ⅲ and Ⅳ, we have re-paste all the figures with high resolution in the revised manuscript.
Figure 1-5 in the revised manuscript.
9) -there is no "discussion" paragraph in which a comparison between the results obtained in this study and comparable literature surveys should be included
Response/Action: Thanks for your suggestion. By incorporating the feedback from reviewer Ⅲ and Ⅳ, we have added the section of "discussion" in the revised edition.
Lines 644-712:
Section of “Discussion” in the revised manuscript.
10) -the bibliographical references must all be re-checked: in many cases not all the authors are indicated, the number 13 is not clear to what it refers, the criteria of the journal must be followed
Response/Action: Thanks for your reminder. According to your suggestions, we have modified the output style and re-checked all the references. The 13th reference has been modified as required by the format.,and all authors have been listed.
Lines 869:
“13. Proposed Designation of Dicyclohexyl Phthalate(CASRN 84-61-7)as a High-Priority Substance for Risk Evaluation.https://www.regulations.gov/document/EPA-HQ-OPPT-2018-0504-0009.2018.”
Sincerely,
Yongning Wu
China National Center for Food Safety Risk Assessment

Reviewer 3 Report
Comments and Suggestions for Authors
The authors presented a manuscript describing how exposure to select phthalic acid esters altered the RNA expression in commercial HepG2 cells and white adipocytes. They exposed different concentrations of PAEs ranging from 200 to 1.6 micromolar which provided an excellent dataset to describe effects based on dosage. There were many references to metabolites of the PAEs but no experiment to understand how the PAE are metabolized by either the hepatocytes or adipocytes. The manuscript would be enhanced by include if the parent of a metabolite of the PAE is responsible for the effect. Authors state that the DEHP and DCHP are rapidly metabolized, but not clear why they first included in the discussion then dropped to focus on MEHP and MCHP. No evidence was included if DEHP and DCHP are metabolize in the adipocytes or hepatocytes. The authors show data supporting metabolic alterations with exposure of MEHP and MCHP, however no toxic effects were included in the study. The study would benefit from additional metabolomic data to support hypotheses. Here are additional suggestions to improve the manuscript:
Revise title to reflect differential disruption of glucose and lipid metabolism, not phthalate metabolites.
There are multiple instances where full sentences are repeated within the manuscript.
Line 121, the use of “gradient concentration” is not clear. How were the PAEs prepared.
The manuscript needs to be clear of number of replicates and exactly what the data represents. For example, the mean stain area is an average on one image experiment or average of multiple replicates.
Line 172 starts an odd sounding paragraph, can be removed.
Quality of the figure needs to be improved, difficult to read label in Figure 1D for example. Figure 1C is not clear if one dose of PAE or multiple. Define MFI. Guiding line for 2x fold change and 0.05 significant in the volcano plot would improve the manuscript.
Line 188, starting with “However, a significant…” is not clear.
Resolution of Figure 2 needs improvement. Appropriate caption should be included.
Line 220 referenced supplementary figure S1A, not include in review materials.
All figures need resolution improvement.
The authors include discussion of changes of expression for specific gene and link together for a hypothesis; however, the manuscript could be greatly improved by including up or down regulation into the hypothesis of alteration of lipid metabolism.
The ethic statement suggest subjects were used in this study but not clear where.
Acknowledge section needs revision.
References the instructive text should be removed.
Comments on the Quality of English LanguageNeeds proofing in a major way.
Author Response
The Referee (III) of
Toxics:
Dear Sir:
We have received your comments on the paper entitled: Differential disruption of phthalate metabolites on glucose and lipid metabolism in human hepatocytes and white adipocytes
The authors presented a manuscript describing how exposure to select phthalic acid esters altered the RNA expression in commercial HepG2 cells and white adipocytes. They exposed different concentrations of PAEs ranging from 200 to 1.6 micromolar which provided an excellent dataset to describe effects based on dosage.
Thank you very much for your helpful suggestions. We appreciate your help for the improvement of the manuscript. We agree with your opinions and we have revised the manuscript according to the suggestions. Followings are our responses to your comments.
1) There were many references to metabolites of the PAEs but no experiment to understand how the PAE are metabolized by either the hepatocytes or adipocytes. The manuscript would be enhanced by include if the parent of a metabolite of the PAE is responsible for the effect. Authors state that the DEHP and DCHP are rapidly metabolized, but not clear why they first included in the discussion then dropped to focus on MEHP and MCHP. No evidence was included if DEHP and DCHP are metabolize in the adipocytes or hepatocytes.
Response/Action: Thank you very much for your helpful suggestions. In the revised manuscript, we have supplemented the metabolism process of PAEs to elucidate why we dropped to focus on MEHP and MCHP.
Lines 310-346:
“ The first step in the metabolism of PAEs is hydrolysis to form monoesters. Hydrolytic enzymes may include several different carboxylesterases, including lipases[5]. PAEs can be rapidly metabolized in the body, with studies showing that pancreatic choles-terol esterases from porcine and bovine can hydrolyze PAEs into monoesters within 15 minutes to 6 hours[20]. Our research indicates that the effects of parent phthalates are lower than their metabolites, leading us to speculate that the harmful effects of PAEs primarily stem from their metabolites.”
2) The authors show data supporting metabolic alterations with exposure of MEHP and MCHP, however no toxic effects were included in the study. The study would benefit from additional metabolomic data to support hypotheses.
Response/Action: Thanks for your suggestion. According to your feedback and that of reviewer I, we have delineated the toxic effects induced by MEHP and MCHP and connected them to our research findings.
Line:706-712:
“Exposure to PAEs can cause multiple adverse health effects. Previous studies have indicated that exposure to PAEs may lead to energy metabolism-related diseases such as non-alcoholic fatty liver disease, atherosclerosis, and diabetes[5,15]. However, the molecular mechanisms underlying these effects remain unclear. Our study suggests that MEHP and MCHP can affect the energy metabolism of hepatic cells and white adipose cells through mechanisms that involve the PPAR and AMPK signaling path-ways, thereby providing insights into the potential metabolic health consequences in-duced by PAEs.”
Here are additional suggestions to improve the manuscript:
3) Revise title to reflect differential disruption of glucose and lipid metabolism, not phthalate metabolites.
Response/Action: Thank you very much for your helpful suggestions. We have revised the title of the article according to your suggestion.
Lines 2-3:
“Differential disruption of glucose and lipid metabolism induced by phthalates in human hepatocytes and white adipocytes”
4) There are multiple instances where full sentences are repeated within the manuscript.
Response/Action: Thanks for this reminding. By incorporating the feedback from reviewer Ⅱ and Ⅳ, we have removed duplicate sentences and carefully checked throughout the manuscript to avoid similar mistakes in the revision.
Lines 167-168:
“A concentration range of 0-200 μM of PAEs was employed, with all concentrations showing no cytotoxicity towards the cells. ”.
Lines 196-264:
“The purified double-stranded cDNA underwent screening, amplification, and purification processes to obtain the final library”.
Lines 269-270:
“Gene expression levels for each sample were quantitatively analyzed using the featureCounts tool in the Subread software”.
Lines 286-289:
“The cells were then incubated with the primary antibody, which was prepared in 1X permeabilization buffer, for 30 minutes at RT. After washing, cells were incubated with fluorescence-conjugated secondary antibodies, also prepared in 1X permeabilization buffer, for 60 minutes. Then the cells were incubated with Hoechst 33342 staining so-lution (Solarbio, Lot: C0031) at room temperature for 5 minutes to label the cell nuclei. Finally, FluoroBriteTM DMEM was used to minimize background fluorescence”.
5) Lines 121, the use of “gradient concentration” is not clear.
Response/Action: Thanks for this reminding. The description of compound concentrations has been modified in the revised edition and the preparation method of PAEs has been supplemented.
Lines 167-168:
“A concentration range of 0-200 μM of PAEs was employed, with all concentrations showing no cytotoxicity towards the cells.”
6) How were the PAEs prepared.
Thanks for this reminding. According to your suggestions, wec have supplemented the PAEs preparation method in the revised manuscript.
Lines 162-165:
“The PAEs were dissolved in Dimethyl sulfoxide (DMSO) to prepare a stock solution of 400 mM, which was stored at -20 ℃. The stock solution was further diluted 1000-fold to prepare a working solution, which was subsequently diluted 2-fold to prepare PAEs ranging from 0 to 200 μM.”
7) The manuscript needs to be clear of number of replicates and exactly what the data represents. For example, the mean stain area is an average on one image experiment or average of multiple replicates.
Response/Action: Thanks for your reminder. We have added the number of replicates and descriptions of mean stained area in the revised manuscript.
Lines 179-180:
“There were a minimum of 2 replicates well for each concentration in each independent experiment, and all data represents at least 3 independent experiments.”
Lines 294-296:
“The mean stained area (MSA) was calculated by normalizing the overall fluorescent staining cells (area and counts) to the total cells in the same image. The mean fluorescence intensity (MFI) was the average pixel intensity over all of the pixels in an spheroid.”
8) Lines 172 starts an odd sounding paragraph, can be removed.
Response/Action: Thanks for your reminder. Line 172 is descriptive phrases of the template. We have removed all templates that appeared in the article and carefully reviewed the entire manuscript to avoid similar mistakes.
9) Quality of the figure needs to be improved, difficult to read label in Figure 1D for example.
Response/Action: This is our mistake of uploading the lower resolution figures in the submission system. According to the suggestions of reviewer II and reviewer Ⅳ, we have re-paste all the figures with high resolution in the revised manuscript. In addition, a more comprehensive figure legend was included in the revised version of the paper.
Figure 1-5 in the revised manuscript.
10) Figure 1C is not clear if one dose of PAE or multiple.
Response/Action: Thanks for your reminder. Figure 1C are representative images of lipid droplets (LDs). HepG2 cell 3D spheroids were treated with various concentrations (0-200) of MEHP and MCHP. Detailed information has been added to the figure legend.
Lines 365-368:
“ Figure 1. The effects of DEHP, DCHP and metabolites on lipid metabolism in HepG2 cells., (C) Representative images of LDs in HepG2 cell 3D spheroids. Dose–response curves(down) were determined by comparing the MFI to that of the control cells. HepG2 cell 3D spheroids were treated with various concentrations(0-200) of MEHP and MCHP for 48h and then incubated with the fluorescent dyes BODIPY 493/503.”
11) Define MFI.
Response/Action: Thanks for your reminder. According to your suggestions, we have defined MFI in the revised manuscript.
Lines 294-296:
“The mean stained area (MSA) was calculated by normalizing the overall fluorescent staining cells (area and counts) to the total cells in the same image. The mean fluores-cence intensity (MFI) was the average pixel intensity over all of the pixels in an spheroid.”
12) Guiding Lines for 2x fold change and 0.05 significant in the volcano plot would improve the manuscript.
Response/Action: Thank you very much for your helpful suggestions. Auxiliary wires have been added as suggested. All the data in the volcano plot had P values less than 0.05.
13) Lines 188, starting with “However, a significant…” is not clear.
Response/Action: Thanks for your reminder. According to your suggestions, we have reorganized the sentences as follows.
Lines 349-351:
“RNA sequencing revealed that MEHP induced a greater number of differentially expressed genes (DGEs) in HepG2 cells than MCHP. And a significant overlap of 23.7% of DGEs was shared between MEHP and MCHP (Figure 1D)”.
14) Resolution of Figure 2 needs improvement. Appropriate caption should be included.
Response/Action: This is our mistake of uploading the lower resolution figures in the submission system. According to the suggestions of reviewer II and reviewer IV, we have re-paste all the figures with high resolution in the revised manuscript. In addition, a more comprehensive figure legend was included in the revised version of the paper.
Lines 436-446:
“Figure 2: Effects of MEHP and MCHP on glucose metabolism and transcriptome in HepG2 cells. (A) GSEA of signaling pathways related to lipid metabolism regulated by MEHP and MCHP; (B) Heat map of DEGs in signaling pathways related to lipid metabolism regulated by MEHP and MCHP; (C) Scatter plot of DEGs in the AMPK signaling pathway under the influence of MEHP and MCHP. HepG2 cells were treated with different concentrations (0-200 μM) of MEHP and MCHP for 72 hours, and the expression levels of relevant proteins were assessed. (D, E, F) Representative im-ages and dose-response curves of the effects of MEHP and MCHP on the protein expression levels of FBP1, G6PC, and CPT1A in HepG2 cells; (G) Schematic representation of the effects of MEHP and MCHP on glucose and lipid metabolism in HepG2 cells via the AMPK, PPAR, and OXPHOS signaling pathways. Red indicates an increase in gene expression, green indicates a decrease in gene expression, and black indicates no statistically significant difference in gene expression.”
15) Lines 220 referenced supplementary figure S1A, not include in review materials.
Response/Action: Thanks for your reminder. Supplementary materials have been uploaded to the submission system
16) All figures need resolution improvement.
Response/Action: This is our mistake of uploading the lower resolution figures in the submission system. According to the suggestions of reviewer II and reviewer Ⅳ, we have re-paste all the figures with high resolution in the revised manuscript. In addition, a more comprehensive figure legend was included in the revised version of the paper.
Figure 1-5 in the revised manuscript.
17) The authors include discussion of changes of expression for specific gene and link together for a hypothesis; however, the manuscript could be greatly improved by including up or down regulation into the hypothesis of alteration of lipid metabolism.
Response/Action: Thank you very much for your helpful suggestions. The changes of genes related to lipid metabolism were identified in Figure 2G.
Figure legend of Figure 2G:
Lines 445-446:
“Red indicates an increase in gene expression, green indicates a decrease in gene expression, and black indicates no statistically significant difference in gene expression.”
18) The ethic statement suggest subjects were used in this study but not clear where.
Response/Action: Thanks for your reminder. According to your suggestions, we have rewritten the ethical statement in the revised manuscript.
Lines 743-747:
“The HepG2 cells was obtained from the Cell Bank of Type Culture Collection Committee of the Chinese Academy of Sciences. White fat progenitor cells of human origin were obtained from the American Type Culture Collection (ATCC).The study was conducted in accordance with the Declaration of Helsinki, and the protocol was ap-proved by the Ethics Committee of China National Center for Food Safety Risk Assessment (Project identification code : 2019004).”
19) Acknowledge section needs revision.
Response/Action: Thanks for your reminder. According to your suggestions, we have revised the acknowledgement.
Lines 753-754:
“Acknowledgments: This work was supported by the National Natural Science Foundation of China (Grant Number 82173564).”
20) References the instructive text should be removed.
Response/Action: Thanks for your reminder. According to your suggestions, we have removed all templates that appeared in the article and carefully reviewed the entire manuscript to avoid similar mistakes.
Sincerely,
Yongning Wu
China National Center for Food Safety Risk Assessment

Reviewer 4 Report
Comments and Suggestions for Authors
Manuscript ID: toxics-2842729
Title: Differential disruption of phthalate metabolites on glucose and lipid metabolism in human hepatocytes and white adipocytes
Summary: This manuscript reports experimental in vitro data from human hepatocellular carcinoma and white adipocyte cell lines exposed to two phthalates and their primary metabolites. The data indicate effects on lipid and carbohydrate metabolism primarily from exposure to the active metabolites.
Comments:
1. The Methods need more detail to fully report the experimental methods in relation to what it is reported in the Results and Discussion section. For example, the Results report data from adipocyte exposure to “Rosi” and “T007”, however these compounds are not reported in the Methods with supplier and purity information like the other compounds used. Further, the authors report that 2 replicate wells were used per concentration, but it is not reported how many independent cell passages or assay plates were used, so it is not clear what the actual sample size is for the data reported. If the assays were all conducted with just two wells per dose level in a single assay plate (single passage), then this is technically n=1 and is not acceptable. Also, in Figure 3 and Results there is discussion of data from pre-adipocytes, differentiating adipocytes, and mature adipocytes but there is no mention of this in the Methods.
2. A major reason for the “differential disruption” reported in the title and throughout is due to the clear difference in potency between MCHP and MEHP. MEHP was a more potent and so produced effects at lower exposure levels. In order to compare the gene expression pathways it would make more sense to evaluate them at equal response levels as opposed to comparing them at the same dose level.
3. Line 82: Define PPAR at first use
4. Line 86: The CAS for DEHP is missing a digit at the end
5. Line 94: Define T3
6. Lines 90-95: Many of the “Lot” numbers reported here are actually “Cat” numbers from Sigma Aldrich. All these need to be checked for accuracy.
7. Line 120: This line says that exposures were 24-72 hours. Were the exposures not a consistent time interval? This is confusing. The cells should have been exposed for a similar amount of time. Please clarify.
8. Lines 120-123 and 123-26 are exact repeats of the same sentence.
9. Line 126-127: The data in the Figures indicates that MEHP and MCHP exposure was conducted using a range of doses. Why here does it state that only a single dose level (200uM) was used. Further, why does it state that MCHP and MEHP treatment was also done “in combination” – I don’t see any data showing combined exposure effect.
10. Lines 130-131, 131-132, and 132-133 are the exact same sentence repeated three times.
11. Lines 142-144, 144-145, and 145-147 are also the exact same sentence repeated three times.
12. Lines 172-174: These lines were just copied out of the author instructions for the journal, delete.
13. Line 175: Delete “Subsection”
14. All Figures: The y-axis labels of “MSA” are not defined in any captions.
15. Figure 2A and 4A: This panel is too small for the reader to read or interpret the figures.
Comments on the Quality of English LanguageEnglish language is acceptable but some minor edits needed. Several sentences were repeated.
Author Response
The Referee (IV) of
Toxics:
Dear Sir:
We have received your comments on the paper entitled: Differential disruption of phthalate metabolites on glucose and lipid metabolism in human hepatocytes and white adipocytes
This manuscript reports experimental in vitro data from human hepatocellular carcinoma and white adipocyte cell Lines exposed to two phthalates and their primary metabolites. The data indicate effects on lipid and carbohydrate metabolism primarily from exposure to the active metabolites.
1) The Methods need more detail to fully report the experimental methods in relation to what it is reported in the Results and Discussion section. For example, the Results report data from adipocyte exposure to “Rosi” and “T007”, however these compounds are not reported in the Methods with supplier and purity information like the other compounds used.
Response/Action: Thanks for your reminder. According to your suggestions, we have supplemented the methods in more detail.
Lines 90-98:
“DEHP (AccuStandard), MEHP (AccuStandard), DCHP (AccuStandard), MCHP (AccuStandard) were purchased from AccuStandard, with a purity of >98%. Calcium pantothenate (Sigma Aldrich), Human insulin (MCE), Biotin (Sigma Aldrich, CAS#: 58-85-5, Cat#:B4639-100mg), Dexamethasone (Sigma Aldrich), Indomethacin (Sigma Aldrich), Isobutylmethylxanthine (Sigma Aldrich), 3,3’,5-Triiodo-L-thyronine (T3, Sigma Aldrich) were purchased from Sigma Aldrich or MCE. Except for calcium pan-tothenate(unlabeled), the purity of reagents mentioned above were greater than 95%. GW6471 (MCE), GSK3787 (MCE), T0070907 (T007, R&D), and Rosi (Cayman) were purchased from MCE, R&D or Cayman. Detail information of the reagents are provided in Table S2.”
2) Further, the authors report that 2 replicate wells were used per concentration, but it is not reported how many independent cell passages or assay plates were used, so it is not clear what the actual sample size is for the data reported. If the assays were all conducted with just two wells per dose level in a single assay plate (single passage), then this is technically n=1 and is not acceptable.
Response/Action: Thanks for your reminder. The number of replicates are clearly stated in the revised manuscript.
Lines 179-180:
“There were a minimum of 2 replicates well for each concentration in each independent experiment, and all data represents at least 3 independent experiments.”
3) Also, in Figure 3 and Results there is discussion of data from pre-adipocytes, differentiating adipocytes, and mature adipocytes but there is no mention of this in the Methods.
Response/Action: Thanks for your reminder. According to your suggestions, we have supplemented the methods in more detail.
Lines 183-190:
“To examine the transcription level changes of adipocytes at different stages, WAT cells (200,000 cells/well) were seeded in 6-well plates (Corning, Lot: 3516) and incu-bated for 24 hours. Subsequently, they were treated with MEHP (100 μM), MCHP (100 μM), Rosi (10 μM), and T007 (1 μM) in replicate wells (n= 3) for 72 hours in complete medium or 12 days in induction medium. After the initial 24-hour culture, the WAT cells were exposed to induction medium (refreshed every 3 days) for 12 days, followed by treatment with MEHP (100 μM), MCHP (100 μM), Rosi (10 μM), and T007 (1 μM) in replicate wells (n= 3) for 72 hours in complete medium.”
4) A major reason for the “differential disruption” reported in the title and throughout is due to the clear difference in potency between MCHP and MEHP. MEHP was a more potent and so produced effects at lower exposure levels. In order to compare the gene expression pathways it would make more sense to evaluate them at equal response levels as opposed to comparing them at the same dose level.
Response/Action: Thank you very much for your helpful suggestions. The disruptive effects of MEHP and MCHP on glucose metabolism and lipid metabolism are different. By calculating the BMDL10, we found that MEHP has a stronger impact on lipid metabolism than MCHP, while MCHP has a stronger impact on glucose metabolism than MEHP.
We performed a transcriptional analysis using a single dose to examine the molecular mechanisms that contribute to the effects of MEHP and MCHP on lipid metabolism.
5) Lines 82: Define PPAR at first use
Response/Action: Thanks for your reminder. We have carefully checked the article to ensure the correct use of acronyms.
Lines 84-86:
“The differential effects between the two cell types are likely due to the varying expres-sion of Peroxisome proliferator-activated receptors (PPARs).”
6) Lines 86: The CAS for DEHP is missing a digit at the end
Response/Action: Thanks for your reminder. The CAS number of DEHP has been modified in the revised version.
Table S1
7) Lines 94: Define T3
Response/Action:Thanks for your reminder. The full name of T3 has been added to the revised version.
Line 94:
“ 3,3’,5-Triiodo-L-thyronine (T3, Sigma Aldrich)”
8) Lines 90-95: Many of the “Lot” numbers reported here are actually “Cat” numbers from Sigma Aldrich. All these need to be checked for accuracy.
Response/Action:Thanks for your reminder. The Lot number of reagents from sigma has been modified to Cat number.
Lines 91-97
“ Calcium pantothenate (Sigma Aldrich), Human insulin (MCE), Biotin (Sigma Aldrich, CAS#: 58-85-5, Cat#:B4639-100mg), Dexamethasone (Sigma Aldrich), Indomethacin (Sigma Aldrich), Isobutylmethylxanthine (Sigma Aldrich), 3,3’,5-Triiodo-L-thyronine (T3, Sigma Aldrich) were purchased from Sigma Aldrich or MCE. Except for calcium pantothenate(unlabeled), the purity of reagents mentioned above were greater than 95%. GW6471 (MCE), GSK3787 (MCE), T0070907 (T007, R&D), and Rosi (Cayman) were purchased from MCE or Cayman”
9) Lines 120: This Lines says that exposures were 24-72 hours. Were the exposures not a consistent time interval? This is confusing. The cells should have been exposed for a similar amount of time. Please clarify.
Response/Action: Thanks for your reminder. We have revised the description in the article where it was not clear.
Lines 176-178:
“ Changes in lipid uptake and accumulation in the cells can be detected as early as 24 hours of treatment. In the context of gene transcription into proteins, a specific duration is required for this process. Consequently, our processing timeframe extended to 72 hours.”
10) Lines 120-123 and 123-26 are exact repeats of the same sentence.
Lines 130-131, 131-132, and 132-133 are the exact same sentence repeated three times.
Lines 142-144, 144-145, and 145-147 are also the exact same sentence repeated three times.
Response/Action: Thanks for this reminding. We have removed duplicate sentences and carefully checked throughout the manuscript to avoid similar mistakes in the revision.
Lines 167-168:
“A concentration range of 0-200 μM of PAEs was employed, with all concentrations showing no cytotoxicity towards the cells. ”.
Lines 196-264:
“The purified double-stranded cDNA underwent screening, amplification, and purification processes to obtain the final library”.
Lines 269-270:
“Gene expression levels for each sample were quantitatively analyzed using the featureCounts tool in the Subread software”.
Lines 286-289:
“The cells were then incubated with the primary antibody, which was prepared in 1X permeabilization buffer, for 30 minutes at RT. After washing, cells were incubated with fluorescence-conjugated secondary antibodies, also prepared in 1X permeabilization buffer, for 60 minutes. Then the cells were incubated with Hoechst 33342 staining so-lution (Solarbio, Lot: C0031) at room temperature for 5 minutes to label the cell nuclei. Finally, FluoroBriteTM DMEM was used to minimize background fluorescence”.
11) Lines 126-127: The data in the Figures indicates that MEHP and MCHP exposure was conducted using a range of doses. Why here does it state that only a single dose level (200 uM) was used.
Response/Action: Thanks for this reminding. A variety of concentrations were utilized to investigate the relationship between dosage and response. As mentioned in the response to suggestion 4), transcription analysis was conducted using a single dose to examine the molecular mechanisms underlying the effects of MEHP and MCHP on lipid metabolism. We have provided a thorough description of the experimental methods in the revised manuscript.
Lines 167-168:
“A concentration range of 0-200 μM of PAEs was employed, with all concentrations showing no cytotoxicity towards the cells.”
Lines 175-176:
“Dose-response experiments for selective PPARs ligands involved various concentrations of GW6471 (0-10 μM), GSK3787 (0-10 μM),and T007 (0-2 μM). ”
Lines 183-190:
“To examine the transcription level changes of adipocytes at different stages, WAT cells (200,000 cells/well) were seeded in 6-well plates (Corning, Lot: 3516) and incu-bated for 24 hours. Subsequently, they were treated with MEHP (100 μM), MCHP (100 μM), Rosi (10 μM), and T007 (1 μM) in replicate wells (n= 3) for 72 hours in complete medium or 12 days in induction medium. After the initial 24-hour culture, the WAT cells were exposed to induction medium (refreshed every 3 days) for 12 days, followed by treatment with MEHP (100 μM), MCHP (100 μM), Rosi (10 μM), and T007 (1 μM) in replicate wells (n= 3) for 72 hours in complete medium.”
Lines 181-183;
“HepG2 cells were treated with MEHP (100 μM) and MCHP (100 μM) for 72 hours (each treatment concentration with a replicate well number of n=4), and cells were collected for transcriptome sequencing.”
12) Further, why does it state that MCHP and MEHP treatment was also done “in combination” – I don’t see any data showing combined exposure effect.
Response/Action: Thanks for this reminding. The descriptions in “Cell Treatment “are inaccurate. We give an accurate description in figure legend of Figure 5.
Lines 640-643:
“ Figure 5. Role of PPARs in the Cell-type specific responses to MEHP and MCHP. (D) Effects of MEHP or MCHP in combination with PPAR inhibitors on the expression of LDs and PLIN2 proteins in HepG2 cells. (E) Effects of MEHP or MCHP in combination with PPAR inhibitors on the expression of LDs and PLIN2 proteins in MA cells.”
13) Lines 172-174: These Lines were just copied out of the author instructions for the journal, delete.
Lines 175: Delete “Subsection”
Response/Action: Thanks for your reminder. We have removed all templates that appeared in the article and carefully reviewed the entire manuscript to avoid similar mistakes.
14) All Figures: The y-axis labels of “MSA” are not defined in any captions.
Response/Action: Thanks for your reminder. According to your suggestions, we have defined the y-axis labels in the revised manuscript.
Lines 293-296:
“The mean stained area (MSA) was calculated by normalizing the overall fluorescent staining cells (area and counts) to the total cells in the same image. The mean fluorescence intensity (MFI) was the average pixel intensity over all of the pixels in an spheroid .”
15) Figure 2A and 4A: This panel is too small for the reader to read or interpret the figures.
Response/Action: This is our mistake of uploading the lower resolution figures in the submission system. According to the suggestions of reviewer II and reviewer III, we have re-paste all the figures with high resolution in the revised manuscript. In addition, a more comprehensive figure legend was included in the revised version of the paper.
Sincerely,
Yongning Wu
China National Center for Food Safety Risk Assessment

Round 2
Reviewer 1 Report
Comments and Suggestions for Authors
Accept in present form
Author Response
Dear Sir:
We have received your comments on the paper entitled: Differential disruption of phthalate metabolites on glucose and lipid metabolism in human hepatocytes and white adipocytes
With the revisions and responses, we trust that we have successfully addressed all the critiques raised by the reviewers. We hope this manuscript is now acceptable for publication in Toxics.
Sincerely yours,
Yongning Wu
China National Center for Food Safety Risk Assessment
wuyongning@cfsa.net.cn
Reviewer 2 Report
Comments and Suggestions for Authors
the text is now more complete and clear
Author Response

(The authors gave the same response as above.)

Reviewer 3 Report
Comments and Suggestions for Authors
Check the use of DEG, several places DGE was included instead.
Comments on the Quality of English Languageproof the manuscript for consistency issues
Author Response
The Referee (III) of
Toxics:
Dear Sir:
We have received your comments on the paper entitled: Differential disruption of phthalate metabolites on glucose and lipid metabolism in human hepatocytes and white adipocytes
Thank you very much for your helpful suggestions. We appreciate your help for the improvement of the manuscript. We agree with your opinions and we have revised the manuscript according to the suggestions. Followings are our responses to your comments.
Check the use of DEG, several places DGE was included instead.
Response/Action: Thanks for this reminding. We have checked the use of DEGs and carefully reviewed the entire manuscript to ensure the consistency of the manuscript.
Line 208、239、268、270
“DEGs”
Sincerely,
Yongning Wu
China National Center for Food Safety Risk Assessment
